# Through-The-Coating Fabrication of Fiber Bragg Grating Relative Humidity Sensors Using Femtosecond Pulse Duration Infrared Lasers and a Phase Mask

Stephen J. Mihailov *, Huimin Ding, Cyril Hnatovsky, Robert B. Walker, Ping Lu and Manny De Silva

Security and Disruptive Technologies Research Center, National Research Council of Canada, 100 Sussex Drive, Ottawa, ON K1A 0R6, Canada; huimin.ding@nrc-cnrc.gc.ca (H.D.); cyril.hnatovsky@nrc-cnrc.gc.ca (C.H.); robert.walker2@nrc-cnrc.gc.ca (R.B.W.); ping.lu@nrc-cnrc.gc.ca (P.L.); manjula.desilva@nrc-cnrc.gc.ca (M.D.S.)
* Correspondence: stephen.mihailov@nrc-cnrc.gc.ca

**Abstract:** Fiber Bragg grating (FBG) relative humidity (RH) sensors are fabricated in commercially available polyimide (PI)-coated optical fibers with diameters of 50 and 125 μm. Infrared (800 nm) femtosecond pulse duration laser pulses and a phase mask are used to inscribe Type-I and Type-II FBGs directly through the protective polyimide coatings of both 50 and 125 μm diameter fibers without typical fiber processing such as hydrogen loading, cryogenic storage, stripping, recoating or annealing. The devices are then evaluated for their performance as humidity sensors. At telecom wavelengths, the 50 μm diameter fiber devices with a 10 μm thick PI coating had a wavelength shift of the Bragg resonance at a constant temperature of 2.7 pm/%RH, whereas the 125 μm diameter fiber devices with a 17 μm thick PI coating had a wavelength shift of 1.8 pm/%RH. The humidity sensors in the 50 μm diameter fiber demonstrated a more rapid response time to small changes in humidity and a weaker hysteresis when compared to the 125 μm diameter fiber devices. No modification to the PI coatings was observed during fabrication. No difference in RH sensitivity was observed for Type-I devices when compared with Type-II devices with the same fiber. The applicability of this approach for fabricating distributed RH sensing arrays with hundreds of sensing elements on a single fiber is discussed.

**Keywords:** optical fiber sensor; relative humidity sensors; fiber Bragg grating; nonlinear optics

## 1. Introduction

The measurement and monitoring of humidity and moisture have widespread importance in many industries such as food processing, packaging and storage, agriculture, pharmaceuticals and healthcare, as well as in commercial and domestic sectors where heating, ventilation, air conditioning and refrigeration (HVACR) are important. Relative humidity (RH) is defined as the ratio of the amount of water vapor present in the atmosphere to the maximum amount that the atmosphere can hold at the existing temperature. It is the most widely used parameter to quantify the amount of water vapor in the environment. Optical fiber as a sensing platform for RH has certain advantages over more conventional electronic hygrometers, namely, their small size, immunity to electromagnetic interference and chemical inertness. Fiber sensors are typically more sensitive and offer a broader range of capabilities tailored for different applications (e.g., colorimetric, point or distributed). Fiber interferometers based on polymer-filled microcavities [1] or using exotic hydrophilic polymer/silica nanoparticle fiber coatings [2] promise to improve sensor response and sensitivity. Very high RH sensitivities have also been reported for fiber Bragg gratings (FBGs) inscribed in microstructured PMMA polymer fibers [3]. A thorough review of optical fiber RH sensors that are based on optical absorption, grating structures of both Bragg and long-period, Fabry–Perot interferometers, modal interferometers, lossy mode

resonances, etc., is presented in [4]. This particular work will focus on RH sensors based on fiber Bragg gratings within polyimide (PI)-coated silica fibers.

FBG-based sensors are optical filters that are photoinduced in the core of the optical fiber using high-powered ultraviolet [5] or infrared femtosecond (fs) pulse duration lasers [6]. They have been shown to be effective optical fiber sensors for direct measurement of temperature and strain [5]. In order to measure other parameters, a transduction layer often needs to be applied to the FBG in order to convert the measurand of interest into a variation of temperature or strain that can be detected by the Bragg grating. In the case of humidity measurements, polyimide (PI) fiber optic coatings were demonstrated by Kronenberg et al. to be effective moisture-sensitive transduction layers when applied to the FBG [7]. The RH detection function is caused by the swelling of the PI layer when exposed to moisture, which induces a strain on the FBG that results in a detectable wavelength shift of the light reflected by the grating. The sensitivity of the devices to RH was dependent on the thickness of the PI coating applied to the FBG. More recently, David et al. were able to enhance the sensitivity of the RH measurement by recoating FBGs that had been chemically etched to reduce their fiber diameters [8], because the force sensitivity of the optical fiber scales inversely with the square of the fiber diameter [9]. In these instances, the fabrication of the sensors is labor-intensive as fiber stripping, etching and recoating are required. These processes also reduce the fiber reliability, making the fiber sensors extremely fragile.

It has been demonstrated that FBGs can be directly inscribed through a PI protective coating on silica fibers using infrared femtosecond irradiation and a phase mask [10–12] or by the point-by-point method [13]. The resulting through-the-coating (TTC) gratings proved to have higher mechanical strength since no fiber stripping or recoating was required. Recently, an FBG was written point-by-point through the PI coating and used as an RH sensor to monitor concrete corrosion in wastewater pipes [14].

In 2020, the femtosecond laser/phase mask approach was optimized to allow for the TTC inscription of FBGs in ultra-thin, 50 µm diameter PI-coated fibers [15] without fiber photosensitization processes such as deuterium loading, which were previously required in [12]. Namely, the spherical aberration caused by the mask substrate was balanced with conical diffraction produced by the mask at a specific distance from the mask, and the chromatic aberration of the acylindrical (i.e., free from spherical aberration) focusing lens in the exposure setup was cancelled out by the chromatic dispersion of the mask. This resulted in a significant sharpening of the line-shaped focus and permitted the direct inscription of an FBG through the polyimide coating of non-sensitized 50 µm diameter fibers.

In this work, the optimized exposure techniques presented in [15] were used to fabricate TTC FBGs in PI-coated 50 and 125 µm diameter optical fibers acquired from Fibercore. By varying the laser pulse energy and number of incident pulses, low scattering loss TTC FBGs, both thermally unstable (Type-I) and thermally stable (Type-II), were created [16] and then evaluated for their responses to RH using a damp heat environmental chamber. These low-loss gratings are ideally suited for distributed sensing arrays where potentially hundreds of grating elements can be concatenated within a single length of fiber.

## 2. Materials and Methods

The classical optical setup for the phase mask inscription of FBGs was used [15]. The source was a Ti-sapphire regenerative amplifier laser system operating at a central wavelength of 800 nm with a Fourier-transform-limited pulse duration of 80 fs (see Figure 1). The linearly polarized fs-beam was expanded approximately 3.5 times along the optical fiber axis and focused through a zeroth-order-nulled holographic phase mask (M) (Ibsen Photonics) with a 1.07 µm pitch using a plano-convex acylindrical lens (CL) with a focal length of 15 mm. The length of the line-focused beam along the x-axis was defined by the aperture (A) of the CL's holder as 15 mm. The linear polarization of the fs-beam was parallel to the mask grooves as the phase mask was optimized to maximize the zero-order suppression for this polarization. The front surfaces of the polyimide-coated fibers from

Fibercore were placed along the line-shaped focus and aligned parallel to the CL. Each fiber was placed at a distance $d \approx 300$ μm away from the phase mask (see Figure 1), where the confocal parameter of the fs-laser focus was the smallest, and thus the peak intensity in the focus was the highest [15]. The position of the focus in the fiber core was aligned by utilizing the techniques of nonlinear photoluminescence microscopy and dark-field microscopy [17].

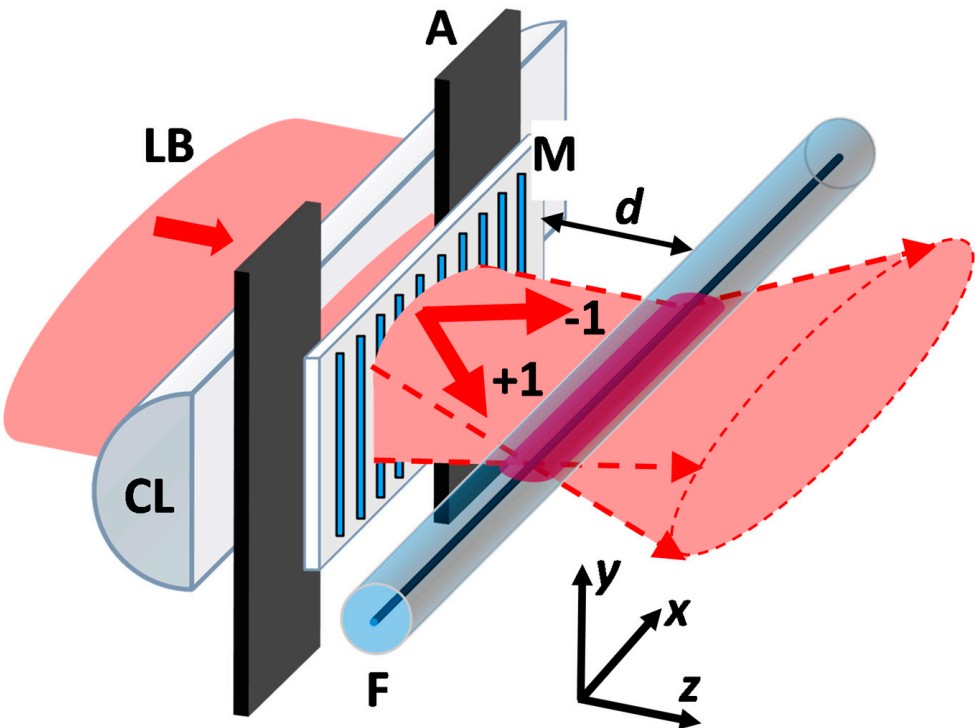

**Figure 1.** Schematic of the laser-writing configuration based on the phase mask technique used to photo-induce changes in the fiber core. The output laser beam ($\approx$7 mm in diameter at the $1/e^2$-intensity level) is expanded along the x-axis. F is the fiber. LB is the femtosecond laser beam; M is the holographic phase mask, which generates only $-1$ and $+1$ diffraction orders at 800 nm; CL is the focusing acylindrical lens; A is the 15 mm wide aperture (along the x-axis) defined by the CL's holder; F is the optical fiber; and $d$ denotes the mask-to-fiber distance (i.e., $\approx$300 μm).

Using this approach, TTC FBGs could be written in both the room temperature stable Type-I regime or the high-temperature stable Type-II regime, depending on the exposure conditions [16]. The specifications of the Ge-doped fibers used and the exposure parameters for the Type-I or Type-II modifications are given in Table 1. The fiber coating thicknesses were measured by comparison of coated and stripped fiber diameters using an optical microscope.

**Table 1.** Fiber parameters and exposure conditions for Type-I and Type-II index changes.

| Fibercore Fiber | Diameter (μm) | PI Coating Thickness (μm) | Type-I Modification Conditions | Type-II Modification Conditions |
|---|---|---|---|---|
| SM1500(7.8/125)P | 125 | 17 | Beam scanning, 1 kHz pulse repetition rate, 195 μJ/pulse | 1 pulse; 850 μJ/pulse |
| SM1500(4.2/50)P | 50 | 10 | No beam scanning, 60 s exposure μat 1 kHz, 95 μJ/pulse | 3–5 pulses; 650 μJ/pulse |

For the 125 µm fiber with a 7.8 µm core diameter and Type-I modification, the acylindrical lens was translated perpendicular to the fiber axis using a piezo-actuated stage in order to scan the beam across the fiber core in a single 20 µm vertical sweep with a duration of 30 s. In all other cases (i.e., Type-I Bragg gratings in the 50 µm diameter fiber and Type-II Bragg gratings in the 50 and 125 µm diameter fibers), the beam focus was centralized within the fiber cores without scanning. The bend-insensitive 50 µm diameter fiber has a smaller core size of 4.2 µm that corresponds to a higher Ge content, making the fibers more photosensitive to femtosecond pulse duration infrared radiation [10]. The pulse energies displayed in Table 1 were measured in front of the phase mask. The fabricated devices were inspected with an optical microscope to determine if there were any visible modifications to the coating surface. None were observed. A total of 12 devices were fabricated: 1 Type-I and 5 Type-II in the PI-coated 125 µm diameter fiber and 4 Type-I and 2 Type-II in the PI-coated 50 µm diameter fiber.

The devices were then tested for their sensitivity to RH using a temperature/humidity test chamber Model MCB(H)-1.2-.33-H/AC from Cincinnati Sub-Zero. The fibers were placed loosely within the chamber with no applied strain. RH levels were varied from 20 to 90%RH at a constant temperature of 40 °C and from 10 to 90%RH at a constant temperature of 60 °C in steps of 10% RH. In both cases, the chamber was programmed to change the RH gradually over 15 min and then to maintain the RH at a stable level for 30 min. RH levels were verified using internal sensors to the unit as well as an Omega HH314A Humidity Temperature meter, which possessed a 0.1%RH resolution and accuracy ± 2.5% RH. The spectral responses from the FBG sensors were monitored using a Micron Optics (Luna) Hyperion FBG interrogator.

### 3. Results

Examples of the transmission and reflection spectra of FBGs written in each of the Fibercore fibers are given in Figure 2. In Figure 2a,b, the transmission and reflection spectra of Type-I TTC gratings fabricated in the 125 µm and 50 µm diameter fibers are denoted by the green and blue traces, respectively. The holographic phase mask with a 1.07 µm pitch typically produces a nominal Bragg resonance wavelength of 1550 nm in standard telecommunication fibers such as SMF-28. In the case of the 125 µm diameter Fibercore fiber, the Bragg resonance wavelength was approximately 1551.4 nm. When written with the same phase mask, the Bragg wavelength of the gratings written in the 50 µm diameter fiber was longer than the Bragg wavelength of the gratings in the 125 µm diameter fiber. Indeed, the core diameter reduction is achieved by increasing the Ge doping levels of the fiber core, which, in turn, increases its effective index and hence creates a Bragg resonance at a longer wavelength for a given grating pitch. The higher Ge doping level associated with core reduction also increases the photosensitivity of the fiber to ultrafast infrared radiation [10]. Figure 2c,d display the transmission and reflection spectra, respectively, of the Type-II TTC FBGs fabricated in the 125 µm and 50 µm diameter fibers, also denoted by the green and blue traces, respectively. It is interesting to note that the appearance of shorter wavelength cladding modes in the transmission spectra, often associated with Type-II ultrafast infrared laser-induced grating writing, are increasingly separated in wavelength from the Bragg resonance as the fiber core size is reduced. This is consistent with what has been observed for FBGs written in high numerical aperture fibers [18].

Both the Type-I and Type-II FBGs fabricated in each of the fiber types were then tested for their responses to changes in RH. The spectral responses of the devices were monitored in reflection using the Hyperion interrogator, which reports a wavelength accuracy of the measurement of 1 pm. As an example, Figure 3 shows the wavelength variation of Type-I TTC FBGs written in the 50 µm diameter fiber and the 125 µm diameter fiber. For comparison, the RH levels as measured by the external Omega meter are also included.

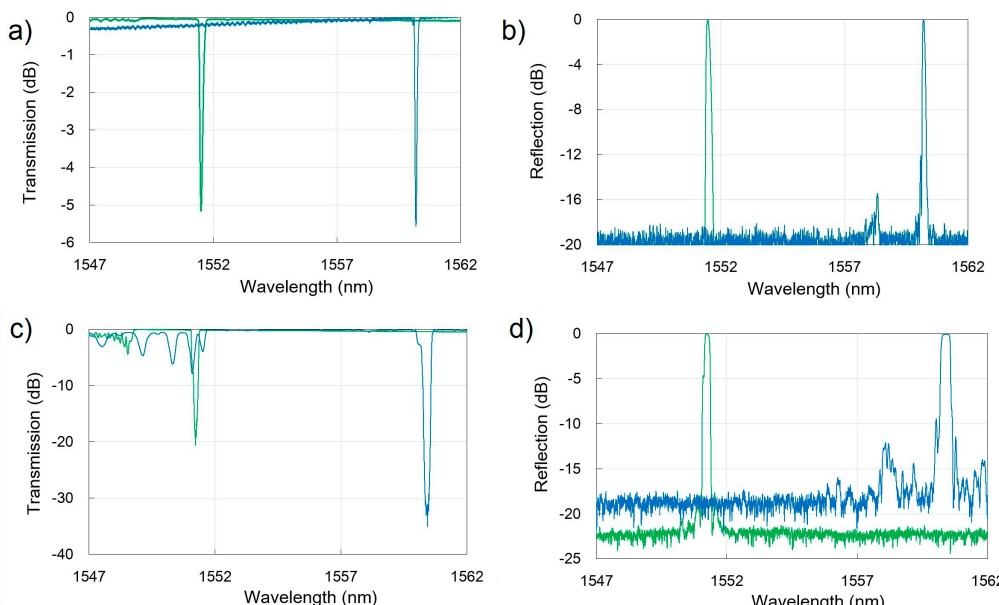

**Figure 2.** Example spectra of TTC FBGs written in PI-coated Fibercore fiber with a 125 μm diameter (green trace) and 50 μm diameter (blue trace). Type-I FBG transmission and reflection spectra are shown in (**a**) and (**b**), respectively, while Type-II FBG transmission and reflection spectra are shown in (**c**) and (**d**), respectively.

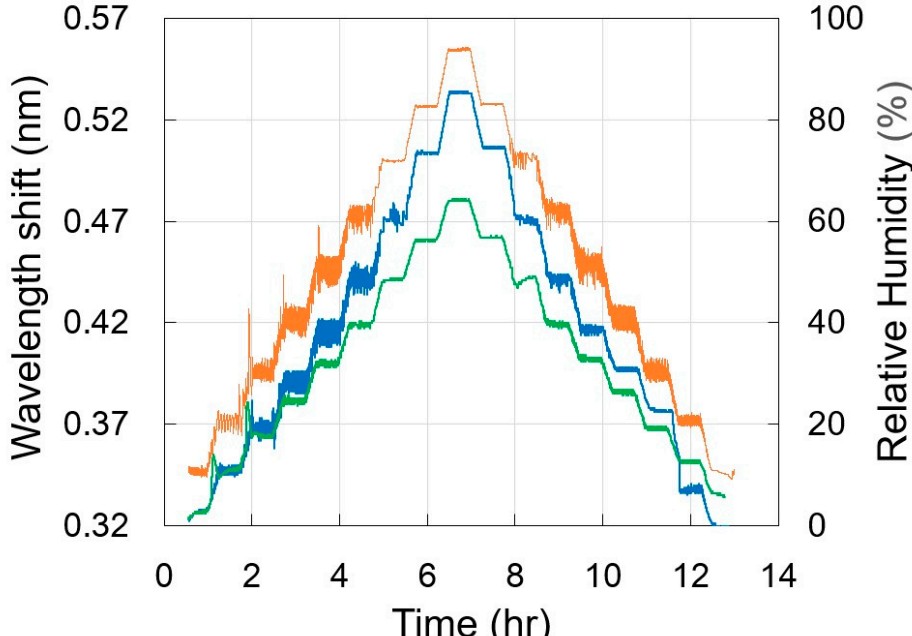

**Figure 3.** Example measurement of TTC FBG wavelength shift and RH as a function of time: the wavelength shift of example FBGs in PI-coated 50 μm and 125 μm diameter fibers are denoted by the blue trace and green trace, respectively. RH measurements were obtained simultaneously with the Omega HH314A Humidity Temperature meter (orange trace). The humidity chamber was maintained at 60 °C in this example. The wavelength shift was normalized to the starting time of the measurement.

Figure 3 shows that below 70% RH, the humidity within the chamber at a given 'fixed' humidity level appears to be noisy with a variation of ± 2.5%RH. A closer examination of the ramp-up to 50%RH shows this oscillation (see Figure 4). There is a direct correlation between the wavelength shift by the FBG RH sensors and the humidity measurement from

the Omega HH314A humidity meter, even on the expanded time scales shown in Figure 4. According to the manufacturer, the response time of the meter is 75 s in slowly moving air. The staircase response of the wavelength shifts of the FBGs is due to the resolution of the FBG interrogator. Because of the reduced thickness of both the coating and the fiber diameter, the 50 μm fiber RH is able to better track variations in RH with time as compared to the sensor fabricated in the 125 μm diameter fiber.

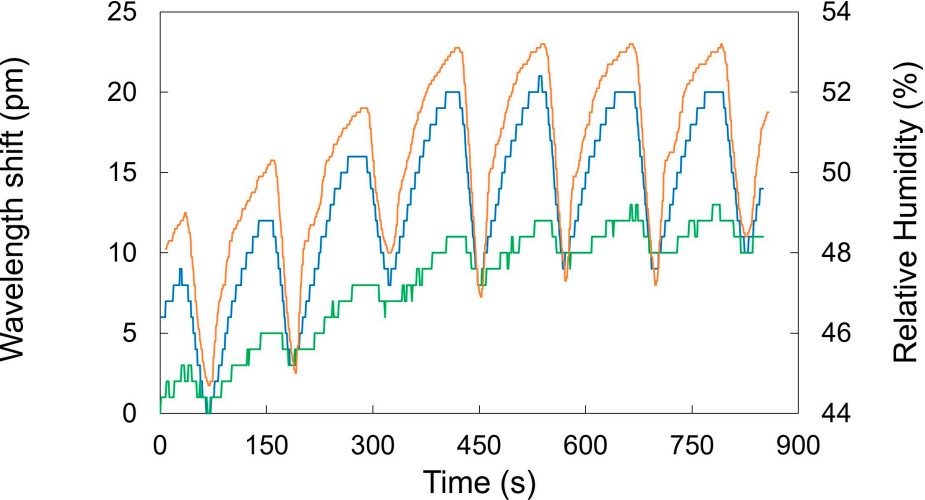

**Figure 4.** Comparison of response times of the 50 μm diameter (blue trace) and 125 μm diameter (green trace) TTC FBGs with changes in relative humidity as measured with the Omega HH314A humidity meter (orange trace).

By re-expressing the kind of data presented in Figure 3, where the wavelength shift for a given constant RH is averaged over 10 min, the variations in wavelength shift versus RH as a function of fiber type and refractive index change type (i.e., Type-I or Type-II) can be obtained. When measured at room temperature, the RH within the humidity chamber was not stable. The wavelength-shift measurements as a function of RH were performed at constant temperatures of 40 °C and 60 °C, as the maximum temperature for the Omega RH meter was limited to 60 °C. An example sensitivity plot of the wavelength shift versus RH of four Type-II devices, two in 50 μm and two in 125 μm diameter fibers, taken at 60 °C is presented in Figure 5. In this instance, the humidity ramp-up and ramp-down were included in the generation of the linear regression trace for each device. As some hysteresis is observable in both Figures 3 and 5, an explicit evaluation of the device hysteresis is presented in Figure 6, where the linear regression traces of data taken during humidity ramp-up and ramp-down are calculated separately. The devices that displayed the largest hysteresis were selected. From the figure, it can be seen that the total hysteresis for the 50 μm and 125 μm diameter devices is denoted by a 13 pm wavelength shift difference when the humidity returns to 10%RH. There was also a slight increase in the device sensitivity on the ramp-down, as denoted by the increase in the linear regression slope.

Figure 7 presents the wavelength shift versus humidity at 60 °C for four 50 μm diameter devices, two written in the Type-I regime and two in the Type-II regime. It can be seen that there is little difference in the response to RH between the Type-I and Type-II devices. There is also little hysteresis in the response of the devices to the ramp-up and ramp-down of the RH. The sensitivity of the devices to RH is defined by the slope of the linear regression of wavelength-versus-RH traces and is presented in Table 2a,b presents the average of the sensitivity values for all devices made with 50 μm diameter fiber (both Type-I and Type-II) and 125 μm diameter fiber at either 40 or 60 °C. The quoted error is the standard deviation of the sensitivity values.

**Table 2.** (**a**). Sensor sensitivities in pm/%RH as a function of temperature and Type-I and Type-II refractive index change and humidity chamber temperature. (**b**). Standard deviation of the device sensitivity response as a function of temperature.

| | | | RH Sensitivity (pm/%RH) | | Regression Slope Linearity $R^2$ (%) | |
|---|---|---|---|---|---|---|
| **Device** | | **FBG Type** | **40 °C** | **60 °C** | **40 °C** | **60 °C** |
| 50 μm | A | I | 2.855 | 2.600 | 99.939 | 98.634 |
| | B | I | 2.731 | 2.548 | 99.800 | 99.169 |
| | C | I | 2.814 | 2.631 | 99.937 | 99.054 |
| | D | I | 2.684 | 2.500 | 99.865 | 99.461 |
| | E | II | 2.700 | 2.793 | 97.728 | 99.240 |
| | F | II | 2.845 | 2.690 | 98.870 | 99.099 |
| 125 μm | A | I | 1.842 | 1.855 | 99.770 | 99.839 |
| | B | II | 1.871 | 1.977 | 99.827 | 99.393 |
| | C | II | 1.827 | 1.767 | 99.515 | 99.765 |
| | D | II | 1.828 | 1.779 | 99.593 | 98.763 |
| | E | II | 1.783 | 1.776 | 99.897 | 99.187 |
| | F | II | 1.858 | 1.776 | 99.645 | 99.286 |

**(b)**

| Device | Temperature | Sensitivity |
|---|---|---|
| 50 μm | 40 °C | 2.77 ± 0.08 pm/%RH |
| | 60 °C | 2.63 ± 0.10 pm/%RH |
| 125 μm | 40 °C | 1.83 ± 0.03 pm/%RH |
| | 60 °C | 1.82 ± 0.08 pm/%RH |

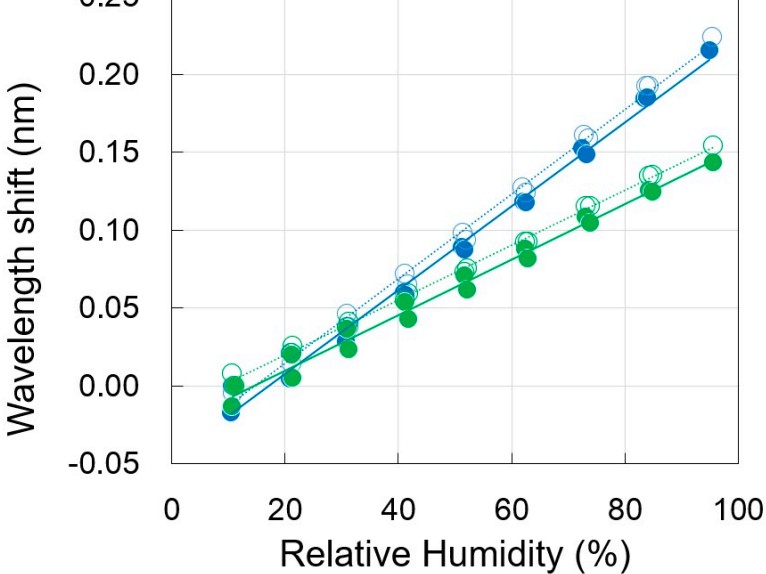

**Figure 5.** Measurements of Type-II TTC FBG wavelength shift as a function of RH at 60 °C for two 125 μm diameter fibers (green traces) and two devices in 50 μm diameter fibers (blue traces). Separate devices with a given fiber diameter are denoted by either open or filled bullet points. Linear regression lines are either solid or dashed for the filled or open bullet points, respectively. Slopes and errors of the regression lines for all of the devices are summarized in Table 2. RH measurements were obtained simultaneously with the Omega HH314A Humidity Temperature meter. Wavelength shifts were normalized to the starting RH value of the measurement, 10% RH.

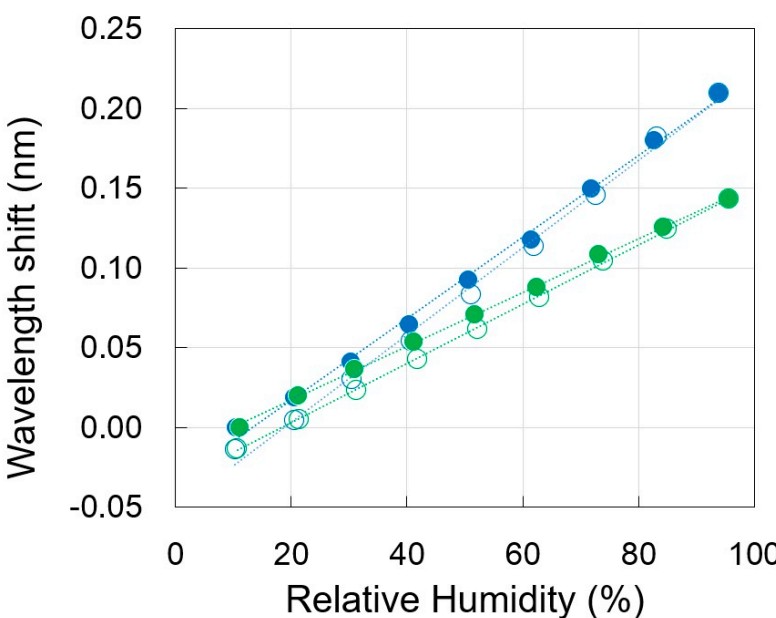

**Figure 6.** Hysteresis measurements at 60 °C for 125 μm (green trace) and 50 μm (blue trace) diameter devices. Solid bullets represent the wavelength shift during the humidity ramp-up, while the open bullets denote the wavelength shift during the humidity ramp-down. The dotted lines represent the linear regression of each ramp-up or ramp-down response.

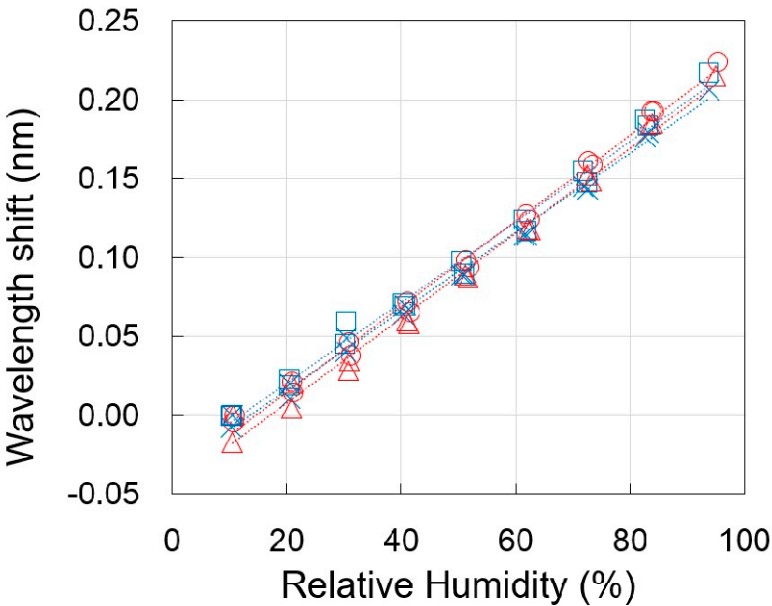

**Figure 7.** 50 μm fiber diameter device sensitivity at 60 °C as a function of index change type. Type-I devices are denoted by blue symbols (cross and square) and Type-II by red symbols (circle and triangle). Both humidity ramp-up and ramp-down points were included in the generation of the regression traces for each device trace.

Measurements on the Type-II devices in 50 μm diameter fiber were repeated on four different occasions at 60 °C, resulting in an average sensitivity of 2.7 ± 0.1 pm/%RH. The resulting sensitivity and error values of the repeated measurements were consistent with the overall average measurement value and error presented in Table 2b. Hysteresis of the devices improved after the fourth cycle from the worst-case wavelength difference of 13 pm shown in Figure 6 to 6 pm or ~2.5% RH, i.e., within error of the reference Omega RH probe.

## 4. Discussion

The response to RH of the TTC Bragg gratings in the PI-coated 125 μm diameter fiber from Fibercore listed in Table 2b is consistent with other standard PI-coated FBGs in the literature, where, for example, PI-coated FBGs from Fiberlogix and Avensys (with PI thicknesses of 14 μm and 18 μm, respectively) had wavelength sensitivities to RH of 1.3 and 1.5 pm/%RH, respectively [14,19]. Similarly, the response of the TTC Bragg gratings in the 50 μm diameter fiber is comparable to that observed for FBG RH sensors in tapered fibers. In that case, the FBGs were written in a 125 μm fiber, after which the fiber was etched to a 54 μm diameter and then coated with 20 μm of PI. The reported sensitivity of the FBG RH sensors in tapered fibers was 3.2 pm/%RH [19]. A comparison of the results from this work with some of those found in the literature is given in Table 3.

**Table 3.** Comparison of state-of-the-art optical fiber-based humidity sensors. (SM: single-mode; MM: multimode; PMMA: Poly(methyl methacrylate); fs PbP: femtosecond laser and the point-by-point technique).

| OFS Structure | Waveguide Diameter (μm) | PI Coating Thickness (μm) | RH Sensitivity | Ref. |
|---|---|---|---|---|
| PI-coated FBG in SM silica fiber | 125 | 29.3 | $2.58 \pm 0.12$ pm/%RH | [7] |
| PI-coated FBG in SM silica fiber and tapers | 125<br>54 | 18<br>19 | 1.5 pm/%RH<br>3.17 pm/%RH | [19] |
| Fs PbP FBG in SM silica fiber | 125 | 14 | 1.3 pm/%RH | [14] |
| Fs phase mask FBG in SM silica fiber | 125<br>50 | 17<br>10 | $1.82 \pm 0.08$ pm/%RH<br>$2.70 \pm 0.10$ pm/%RH | This work |
| Polymer filled F-P cavity | 125 | - | $17.1 \pm 1.7$ pm/%RH | [1] |
| MM fibre tip with hydrophillic polymer/silica nanoparticle coating | 125 | - | Transmittance variation 0.43%/RH% | [2] |
| Microstructured PMMA polymer fiber with FBG | 125 | - | $35 \pm 3$ pm/%RH | [3] |

The important result to note in this work is that the sensors were produced by processing fibers as received from the manufacturer. Inscription of TTC gratings in PI-coated 125, 80 and 50 μm fibers using femtosecond lasers and a phase mask was reported by Habel et al. [12]. In that work, only Type-I FBGs were created in the PI-coated 50 μm fiber and only after the fiber had undergone the deuterium-loading fiber photosensitization process [20]. In our case, no fiber processing such as fiber photosensitization, cryogenic storage, stripping, etching, recoating or annealing was required to produce Type-I or Type-II gratings in the 50 μm diameter fiber in order to make effective RH sensors. The higher intensity irradiation required for Type-II grating formation could potentially damage the coating, causing delamination that would affect the sensitivity or performance of the device. No noticeable modification of the PI coating was observed for the FBG sensors written in either the Type-I or Type-II regimes. From Figure 7, it is clear that very similar sensitivities to RH were observed for both Type-I and Type-II sensors in the same fiber. Type-II sensors would be appropriate for measurement of moisture content in flue or exhaust gases at temperatures up to 300 °C, the rated maximum temperature for PI.

TTC inscription of FBGs lends itself to automation of the fabrication of distributed sensing arrays because extremely low scattering loss gratings can be produced in both Type-I and Type-II regimes using the phase mask method [21]. While Type-II devices do

exhibit cladding-mode coupling that is potentially quite high if the gratings have high reflectivity (5 dB in the case of Figure 2c), if interrogated correctly, the lowest wavelength device first, array spacing and intensity loss can be minimized. Type-I devices do not suffer from this constraint because they have very little cladding-mode coupling. As these gratings have extremely low scattering loss, even Type-II, many can be concatenated together. To demonstrate this principle, 160 low-reflectivity Type-II gratings were inscribed through the PI coating of 125 μm telecom fibers using a phase mask array with discrete mask elements to produce sensor elements resonant in the telecommunication S-, C- and L-bands. See Figure 8. These devices were not tested for their response to humidity at the time of publication.

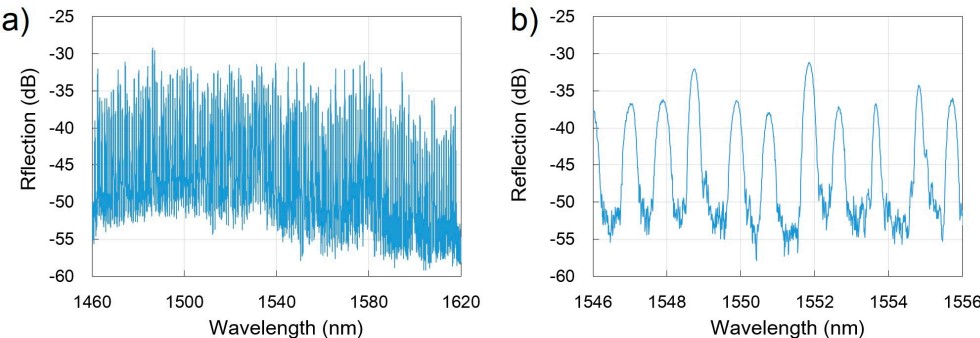

**Figure 8.** Reflection spectra of a 160 element TTC FBG array inscribed in PI-coated 125 μm telecom fibers: (**a**) presents the entire device spectrum over the S-, C- and L-bands, while (**b**) presents a 10 nm wavelength span centered at 1551 nm. Spectral responses were measured with a Hyperion FBG interrogator.

With proper interrogation, either using wide bandwidth FBG interrogator sources or an interrogator that combines wavelength division multiplexing and time division multiplexing (WDM/TDM) in its measurement methodology, effective distributed RH sensing systems could be created. Such a distributed sensor system is of interest to the oil and gas sector, where monitoring of RH and water vapor condensation is important for corrosion detection. With its potential low cost per unit length, simple preparation, easy operation and good sensitivity, a distributed RH sensor system based on TTC FBGs would be a good candidate for a distributed corrosion sensing system for natural gas transmission pipelines [22].

## 5. Conclusions

In this paper, we have shown that sensitive relative humidity sensors can be created in commercially available polyimide-coated 50 μm diameter silica optical fibers by using through-the-coating fiber Bragg grating inscription techniques based on high-powered near-infrared femtosecond pulse duration laser exposure through a phase mask. Devices stable at room temperature (Type-I) or high temperatures (Type-II) showed no difference in RH sensitivity for a given fiber geometry. The devices perform similarly to those published in the literature but are much more easily manufactured as no fiber processing techniques such as hydrogen loading, cryogenic storage, stripping, recoating or annealing are required. With the removal of these several labor-intensive processing steps that degrade fiber reliability, the described technique can be used to fabricate many robust sensors in a single fiber, potentially producing a distributed RH sensor.

**Author Contributions:** Conceptualization, H.D., P.L. and S.J.M.; methodology, C.H. and R.B.W.; software, R.B.W.; validation, H.D.; formal analysis, H.D. and S.J.M.; investigation, H.D.; resources, M.D.S. and R.B.W.; data curation, H.D.; writing—original draft preparation, H.D. and S.J.M.; writing— review and editing, S.J.M.; visualization, S.J.M.; supervision, S.J.M.; project administration, H.D. All authors have read and agreed to the published version of the manuscript.

**Funding:** This research received no external funding.

**Data Availability Statement:** The data presented in this study are available on request from the corresponding author.

**Conflicts of Interest:** The authors declare no conflict of interest.

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
