# Peer review of "Through-The-Coating Fabrication of Fiber Bragg Grating Relative Humidity Sensors Using Femtosecond Pulse Duration Infrared Lasers and a Phase Mask"

_photonics, doi:10.3390/photonics10060625_

Round 1
Reviewer 1 Report
No.
Author Response
We would like to thank the reviewer for their supportive review. We have introduced 3 new references to the introduction to improve the background of the field
Reviewer 2 Report
The article titled ‘Through-the-coating fabrication of fiber Bragg grating relative humidity sensors using femtosecond pulse duration infrared lasers and a phase mask’ shows the effects of humidity on optical fibres of varying thickness coated with polyimide. Fibre optic FBG humidity sensors relying on transducer polymer coatings were first demonstrated more than 20 years ago. Hence, a vast number of papers exist in the literature that cover these devices in depth. In fact, the references cited by the authors themselves cover the results presented in this manuscript. The main conclusions presented by the authors are summarised in lines 236 and 241, namely that their devices compare with those already demonstrated in the literature using a technique the authors published themselves in 2008 (ref 7). The authors do not present any new material (50µm core RH sensor compared to 80µm or 125µm is not compelling), there are no longevity tests, the testing climate chamber is quite noisy and inaccurate (which raises concern regarding the FBG calibrations), and there are no out-of-the-lab tests. In my opinion, the authors do not make a convincing case for novelty or field advancement and I believe the manuscript is not suitable to be published in MDPI Photonics. I recommend rejecting the manuscript.
* Line 111: The authors suggest that a smaller core with higher Ge content makes the fibre more photosensitive. To what? The fabrication wavelength? If so, by how much when comparing IR to UV absorption?
* Lines 130, 153: Transmission spectra are shown, however, reflection peaks were monitored. The authors should also include reflection spectra.
* Lines 133-135: What was the target wavelength?
* Fig 3: These plots are crucial to show how good the fabricated devices are, yet the authors present two small graphs with no grids. I suggest the authors combine the two plots into one plot, make it larger and add a grid. Regardless, the reader can clearly see that all the devices have a very large random hysteresis (contrary to that stated in line 188), and that the climate chamber really struggles to maintain its required levels. This is echoed in Fig 5.
* Fig 5: Previous demonstrations in the literature show that polyimide follows a linear calibration curve. The authors need to fit a linear regression to all the data points rather than just joining them up sequentially. That way the reader will clearly see that the RH sensors have an inherent large and random hysteresis. Furthermore, a grid needs to be added to aid the reader.
* Table 2: How were the error values calculated? Were multiple calibration runs conducted? Can the authors comment on why the sensitivity (pm/%RH) decreases when changing calibration from 40C to 60C for a 50µm core fibre but increases for a 125µm fibre? How does this compare to the literature?
* Lines 212-213: The authors state that an important result in this work is that fibres are processed direct from the manufacturer without stripping and recoating etc. I agree that this is important; however, this is not a new result. See cited refs 7, 9, 10 for example. Moreover, the authors crucially failed to cite J. Habel, T. et al., "Femtosecond FBG Written through the Coating for Sensing Applications," Sensors 17(11), 2519 (2017). Habel et al. directly fabricated FBGs through manufactured polyimide coatings, using a phase mask and IR wavelength, of 50µm, 80µm, and 125µm clad fibres. The work done by Habel et al. nullifies the conclusion (lines 233-241) presented by the current authors.
* Lines 219-220: In my opinion, the FBGs outlined in this manuscript do not lend themselves well to distributed sensing arrays. The 5dB cladding mode losses (Fig 3b) would severely affect the array spacing and intensity of further cascaded FBGs down the line.
Author Response
Responses to Reviewer 2's comments are enclosed in the attached file.

Reviewer 3 Report
This paper reports an interesting study about coating fabrication of FBG relative humidity sensors using femtosecond pulse duration infrared lasers and a phase mask. Some comments.
1. Introduction needs some improvements about the literature of other sensors based on other configurate like cavities or reflective-end for humidity measurement like: Optical and Quantum Electronics 48, 1-8, 2016; Sensors and Actuators B: Chemical, 344, 2021, 130154.
2. How about the reproducibility of the performance using identical probes? the same pro repeatability.
3. why this technology and not other like using polymer optical fibers that presente features to have afinity to humidity? Please comment.
4. A tabular form to present the comparison with state of the art and your work is needed. Consider to present in that table the cost of fabrication (high, low, etc) in terms of fabrication taking into account the laser, instrumentation, etc to produce the sensors. Also compare with intensity based sensor using polymer fiber for such proposes like: Sensors 18 (3), 916, 2018.
Author Response
Responses to Reviewer 3's comments are in the attached file

Round 2
Reviewer 2 Report
I thank the authors for resubmitting a modified article. Whilst most concerns were addressed by the authors, I still feel the results presented do not increment the RH sensor research field. Type I AND Type II FBGs through a PI coating have previously been reported by multiple research groups. Some were assessed for their use as RH sensors, others not. All such previous Type II FBGs were done so in virgin fibre direct from a commercial supplier.
Therefore, the only novel content in this manuscript would be a Type II FBG in a 50µm clad PI coated fibre. The obvious question still remains, does a 50µm clad FBG RH sensor present a great improvement over an identical RH sensor fabricated in 80µm or 125µm clad PI coated fibre? Is a hysteresis of up to 10pm, meaning RH can only be accurate to within 5-6%, acceptable for field use? Are the RH sensors precise? I.e. if a RH sensor was run through the same RH calibration steps continuously over and over again, would the hysteresis keep growing or stablilise?
Author Response
Please see the attachment. Reviewer 2's comments are in black, our responses are in red and the modification to the manuscript is in blue.

Reviewer 3 Report
The paper is ready for publication
Author Response
We thank the reviewer for accepting our first revision of the manuscript